# Imposing Category Trees Onto Word-Embeddings Using A Geometric Construction

**Tiansi Dong[1] & Christian Bauckhage[1,2]**
[1]B-IT, University of Bonn, Bonn, Germany
[2]Fraunhofer IAIS, Sankt Augustin, Germany
{dongt, bauckhag}@bit.uni-bonn.de

**Hailong Jin & Juanzi Li**
DCST, Tsinghua University, Beijing, China
jinhl15@mails.tsinghua.edu.cn
lijuanzi@tsinghua.edu.cn

**Olaf H. Cremers, Daniel Speicher & Armin B. Cremers**
B-IT, University of Bonn, Bonn, Germany
{cremerso, dsp, abc}@bit.uni-bonn.de

**Jörg Zimmermann[1,3]**
[3]Informatik II, University of Bonn
jg@iai.uni-bonn.de

## Abstract

We present a novel method to precisely impose tree-structured category information onto word-embeddings, resulting in ball embeddings in higher dimensional spaces ($\mathcal{N}$-balls for short). Inclusion relations among $\mathcal{N}$-balls implicitly encode subordinate relations among categories. The similarity measurement in terms of the cosine function is enriched by category information. Using a geometric construction method instead of back-propagation, we create large $\mathcal{N}$-ball embeddings that satisfy two conditions: (1) category trees are precisely imposed onto word embeddings at zero energy cost; (2) pre-trained word embeddings are well preserved. A new benchmark data set is created for validating the category of unknown words. Experiments show that $\mathcal{N}$-ball embeddings, carrying category information, significantly outperform word embeddings in the test of nearest neighborhoods, and demonstrate surprisingly good performance in validating categories of unknown words. Source codes and data-sets are free for public access https://github.com/GnodIsNait/nball4tree.git and https://github.com/GnodIsNait/bp94nball.git.

## 1 Introduction

Words in similar contexts have similar semantic and syntactic information. Word embeddings are vector representations of words that reflect this characteristic (Mikolov et al., 2013; Pennington et al., 2014) and have been widely used in AI applications such as question-answering (Tellex et al., 2003), text classification (Sebastiani, 2002), information retrieval (Manning et al., 2008), or even as a building-block for a unified NLP system to process common NLP tasks (Collobert et al., 2011). To enhance semantic reasoning, researchers proposed to represent words in terms of regions instead of vectors. For example, Erk (2009) extended a word vector into a region by estimating the log-linear probability of weighted feature distances and found that hyponym regions often do not fall inside of their hypernym regions. By using external hyponym relations, she obtained 95.2% precision and 43.4% recall in hypernym prediction for a small scale data set. Her experiments suggest that regions structured by hyponym relations may not be located within the same dimension as the space of word embeddings. Yet, how to construct strict inclusion relations among regions is still an open problem when representing hypernym relations.

In this paper, we restrict regions to be $n$ dimensional balls ($\mathcal{N}$-ball for short) and propose a novel geometrical construction approach to impose tree-structured category information onto word embeddings. This is guided by two criteria: (1) Subordinate relations among categories shall be implicitly and precisely represented by inclusion relations among corresponding $\mathcal{N}$-balls. This way, the energy costs of imposing structure will be zero; (2) Pre-trained word embeddings shall be well-preserved. Our particular contributions are as follows: (1) The proposed novel geometric approach achieves zero energy costs of imposing tree structures onto word-embeddings. (2) By considering category information in terms of the boundary of an $\mathcal{N}$-ball, we propose a new similarity measurement that is

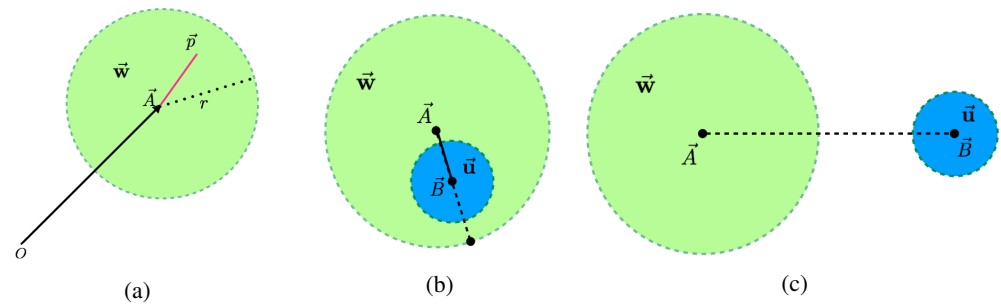

Figure 1: (a) The structure of $\mathcal{N}$-ball; (b) $\overrightarrow{\mathbf{u}}$ is inside $\overrightarrow{\mathbf{w}}$; (c) $\overrightarrow{\mathbf{u}}$ disconnects from $\overrightarrow{\mathbf{w}}$

more precise and sensitive than conventional cosine similarity measurements. (3) We create a large data set of $\mathcal{N}$-ball embeddings using the pre-trained GloVe embeddings and a large category tree of word senses extracted from Word-Net 3.0.

The remainder of our presentation is structured as follows: Section 2 presents the structure of $\mathcal{N}$-ball embeddings; Section 3 describes the geometric approach to construct $\mathcal{N}$-ball embeddings; Section 4 presents experiment results; Section 5 briefly reviews related work; Section 6 concludes the presented work, and lists on-going research.

## 2 REGION-BASED EMBEDDINGS

Tree structures occur in many applications and are used to describe, say, file systems, syntactic structures, taxonomies of plants or animals, or subordinate relations of governments. Here, we use the tree structure of hyponym relations among word senses as to construct $\mathcal{N}$-ball embeddings. Formally, an $\mathcal{N}$-ball of word sense $w$ with central point $\vec{A}_w$ and radius $r_w$ is written as $\overrightarrow{\mathbf{w}} = \mathbb{B}(\vec{A}_w, r_w)$ and defined as the set of vectors whose Euclidean distance to $\vec{A}_w$ is less than $r_w$: $\mathbb{B}(\vec{A}_w, r_w) \triangleq \{\vec{p} | \|\vec{A}_w - \vec{p}\| < r_w\}$. $\mathcal{N}$-balls are defined as open-regions, as illustrated in Figure 1(a), so they are not RCC regions that can be either open or closed, or even a mixture, thus avoiding a number of problems (Dong, 2008; Davis & Marcus, 2015).

### 2.1 RELATIONS BETWEEN $\mathcal{N}$-BALLS

We distinguish two topological relations between $\mathcal{N}$-balls: *being inside* and *being disconnected from* as illustrated in Figure 1(b, c). $\overrightarrow{\mathbf{u}} = \mathbb{B}(\vec{A}_u, r_u)$ being inside $\overrightarrow{\mathbf{w}} = \mathbb{B}(\vec{A}_w, r_w)$ can be measured by the result of subtracting the sum of radius $r_u$ and the distance between their central vectors from radius $r_w$. Formally, we define $\mathcal{D}_{inside}(\overrightarrow{\mathbf{u}}, \overrightarrow{\mathbf{w}}) \triangleq r_w - r_u - \|\vec{A}_u - \vec{A}_w\|$. So, $\overrightarrow{\mathbf{u}}$ is inside of $\overrightarrow{\mathbf{w}}$ ($w$ is the hypernym of $u$), if and only if $\mathcal{D}_{inside}(\overrightarrow{\mathbf{u}}, \overrightarrow{\mathbf{w}}) \geq 0$. If $v$ is the hypernym of $w$, $w$ is the hypernym of $u$, we have $\mathcal{D}_{inside}(\overrightarrow{\mathbf{u}}, \overrightarrow{\mathbf{v}}) > \mathcal{D}_{inside}(\overrightarrow{\mathbf{u}}, \overrightarrow{\mathbf{w}}) + \mathcal{D}_{inside}(\overrightarrow{\mathbf{w}}, \overrightarrow{\mathbf{v}}) \geq \mathcal{D}_{inside}(\overrightarrow{\mathbf{w}}, \overrightarrow{\mathbf{v}})$. The direct hypernym of $\overrightarrow{\mathbf{v}}$, written as $\mathbf{dh}(\overrightarrow{\mathbf{v}})$, can be defined as the $\overrightarrow{\mathbf{x}}$ which produces the minimal positive value of $\mathcal{D}_{inside}(\overrightarrow{\mathbf{v}}, \overrightarrow{\mathbf{x}})$. Formally, $\mathbf{dh}(\overrightarrow{\mathbf{v}}) \triangleq \arg\min_{\overrightarrow{\mathbf{x}}, \overrightarrow{\mathbf{x}} \neq \overrightarrow{\mathbf{v}}} \mathcal{D}_{inside}(\overrightarrow{\mathbf{v}}, \overrightarrow{\mathbf{x}}) \geq 0 = \min\{\overrightarrow{\mathbf{x}} | \mathcal{D}_{inside}(\overrightarrow{\mathbf{v}}, \overrightarrow{\mathbf{x}}) \geq 0 \wedge \overrightarrow{\mathbf{x}} \neq \overrightarrow{\mathbf{v}}\}$.

Similarly, $\overrightarrow{\mathbf{u}}$ disconnecting from $\overrightarrow{\mathbf{w}}$ can be measured by the result of subtracting the distance between their center vectors from the sum of their radii. Formally, we define $\mathcal{D}_{disc}(\overrightarrow{\mathbf{u}}, \overrightarrow{\mathbf{w}}) \triangleq r_w + r_u - \|\vec{A}_u - \vec{A}_w\|$, That is, $\overrightarrow{\mathbf{u}}$ disconnects from $\overrightarrow{\mathbf{w}}$, if and only if $\mathcal{D}_{disc}(\overrightarrow{\mathbf{u}}, \overrightarrow{\mathbf{w}}) \leq 0$.

### 2.2 SIMILARITY MEASUREMENT

Similarity is normally measured by the cosine value of two vectors, e.g., Mikolov et al. (2013). For $\mathcal{N}$-balls, the similarity between two balls can be approximated by the cosine value of their central vectors. Formally, given two $\mathcal{N}$-balls $\overrightarrow{\mathbf{u}} = \mathbb{B}(\vec{A}_u, r_u)$ and $\overrightarrow{\mathbf{w}} = \mathbb{B}(\vec{A}_u, r_w)$, their cosine similarity

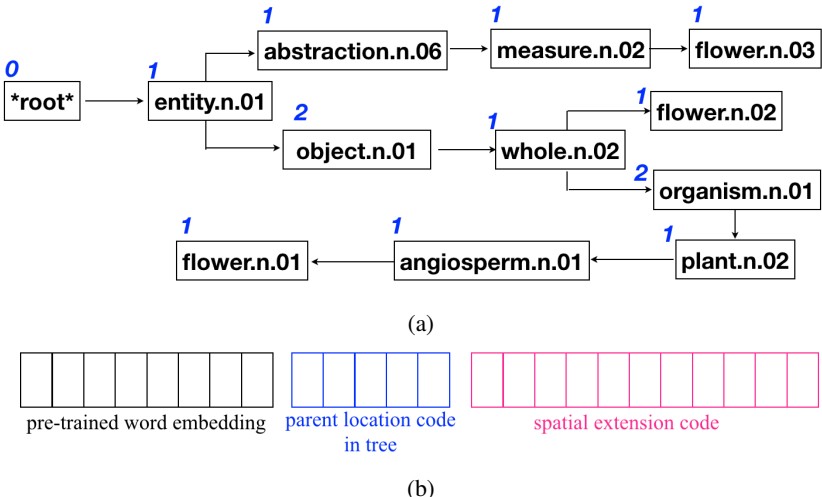

(a)

pre-trained word embedding | parent location code in tree | spatial extension code

(b)

Figure 2: (a) Hypernym relations of three word senses of flower in Word-Net 3.0; (b) Three components of the center point of an $\mathcal{N}$-ball.

can be defined as $\cos(\vec{A}_u, \vec{A}_w)$. One weakness of the method is that we do not know the boundary of the lowest $\cos$ value below which two word senses are not similar. Using category information, we can define that two word senses are not similar, if they have different direct hypernyms. Formally,

$$Sim_0(\overrightarrow{\mathbf{u}}, \overrightarrow{\mathbf{w}}) \triangleq \begin{cases} \cos(\vec{A}_u, \vec{A}_w) & \mathbf{dh}(\overrightarrow{\mathbf{u}}) = \mathbf{dh}(\overrightarrow{\mathbf{w}}) \\ -1 & \text{otherwise} \end{cases}$$

## 3 Constructing $\mathcal{N}$-ball Embeddings

$\mathcal{N}$-ball embeddings encode two types of information: (1) word embeddings and (2) tree structures of hyponym relations among word senses. A word can have several word senses. We need to create a unique vector to describe the location of a word sense in hypernym trees. We introduce a virtual root (*root*) to be the parent of all tree roots, fix word senses in alphabetic order in each layer, and number each word sense based on the fixed order. A fragment tree structure of word senses is illustrated in Figure 2(a). The path of a word sense to *root* can be uniquely identified by the numbers along the path. For example, the path from *root* to *flower.n.03* is [*entity.n.01, abstraction.n.06, measure.n.02, flower.n.03*][1], which can be uniquely named as [1,1,1,1]; We call this vector *location code* of *flower.n.03*. The *location code* of the direct hypernym of *flower.n.03* is called *Parent Location Code (PLC)*. PLC of *flower.n.03* is [1,1,1], PLC of *whole.n.02* is [1,2].

As a word and its hypernym may not co-occur in the same context, their co-occurrence relations can be weak, and the cosine similarity of their word embeddings could even be less than zero. For example, in GloVe embeddings, $\cos(\mathsf{ice\_cream}, \mathsf{dessert}) = -0.1998$, $\cos(\mathsf{tuberose}, \mathsf{plant}) = -0.2191$. It follows that the hypernym ball must contain the origin point of the $n$-dimensional space, if this hypernym ball contains its hyponym, as illustrated in Figure 3 (a). When this happens to two semantically unrelated hypernyms, $\mathcal{N}$-balls of the two unrelated hypernyms shall partially overlap, as they both contain the original point. For example, the ball of *desert* shall partially overlap with the ball of *plant*. This violates our first criterion. To avoid such cases, we require that no $\mathcal{N}$-ball shall contain the origin $O$. This can be computationally achieved by adding dimensions and realized by introducing a *spatial extension code*, a constant non-zero vector, as illustrated in Figure 2(b). To intuitively understand this, imagine that you stand in the middle of two objects A and B, and cannot see both of them. To see both of them without turning the head or the eyes, you would have to walk several steps away so that the angle between A and B is less than some degree, as illustrated in Figure 3(c).

---

[1]We exclude *root* from the path

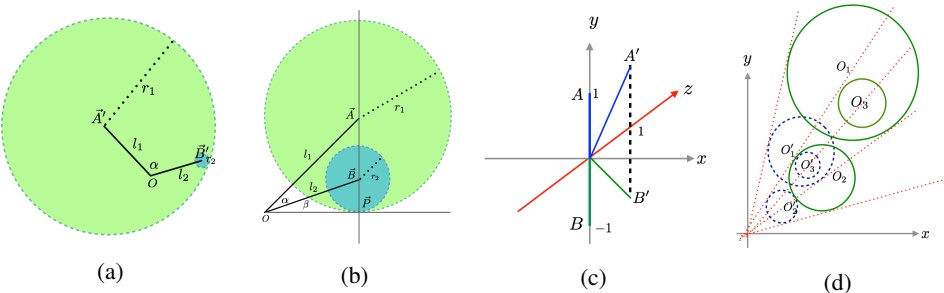

Figure 3: (a) The angle between ball $(\vec{A}', r_1)$ and ball $(\vec{B}', r_2)$ is greater than $90°$, so, ball $(\vec{A}', r_1)$ will contain $O$, if it contains ball $(\vec{B}', r_2)$; (b) If ball $(\vec{A}, r_1)$ contains ball $(\vec{B}, r_2)$, not containing $O$, the sum of $\alpha$ and $\beta$ is less than $90°$; (c) The angle $\angle AOB = 180°$. If we shift A and B one unit into the new dimension, we have $A'(0, 1, 1)$ and $B'(0, -1, 1)$, the angle $\angle A'OB'$ will become $90°$; (d) $O_1, O_2, O_3$ are homothetic to $O_1', O_2', O_3'$, inclusion relations among them are preserved.

## 3.1 THE STRUCTURE OF THE CENTRAL VECTOR

Following Zeng et al. (2014) and Han et al. (2016), we structure the central vector of an $\mathcal{N}$-ball by concatenating three vectors: (1) the pre-trained word-embedding, (2) the PLC (if the code is shorter than the max length, we append 0s till it reaches the fixed length), (3) the spatial extension code.

## 3.2 ZERO ENERGY LOSS FOR $\mathcal{N}$-BALL CONSTRUCTION USING GEOMETRIC APPROACH

Our first criterion is to precisely encode subordinate relations among categories into inclusion relations among $\mathcal{N}$-balls. This is a considerable challenge, as the widely adopted back-propagation training process (Rumelhart et al., 1988) quickly reaches non-zero local minima and terminates[2]. The problem is that when the location or the size of an $\mathcal{N}$-ball is updated to improve its relation with a second ball, its relation to a third ball will deteriorate very easily. We propose the classic depth-first recursion process, listed in **Algorithm 1**, to traverse the category tree and update sizes and locations of $\mathcal{N}$-balls using three geometric transformations as follows.

**Homothetic transformation (H-tran)** which keeps the direction of the central vector and enlarges lengths of $\vec{A}$ and $r$ with the same rate $k$. $\mathcal{H}(\mathbb{B}(\vec{A}, r), k) \triangleq \mathbb{B}(\vec{A}', r')$, satisfying (1) $\frac{|\vec{A}|}{|\vec{A}'|} = \frac{r}{r'} = k > 0$, and (2) $\frac{\vec{A}}{|\vec{A}|} = \frac{\vec{A}'}{|\vec{A}'|}$;

**Shift transformation (S-tran)** which keeps the length of the radius $r$ and adds a new vector $\vec{s}$ to $\vec{A}$. $\mathcal{S}(\mathbb{B}(\vec{A}, r), \vec{s}) \triangleq \mathbb{B}(\vec{A}', r)$, satisfying $\vec{A}' = \vec{A} + \vec{s}$;

**Rotation transformation (R-tran)** which keeps the length of the radius $r$ and rotates angle $\alpha$ of $\vec{A}$ inside the plane spanned by the $i$-th and the $j$-th dimensions of $\vec{A}$. $\mathcal{R}(\mathbb{B}(\vec{A}, r), \alpha, i, j) \triangleq \mathbb{B}(\vec{A}', r)$, such that $\vec{A}_k = \vec{A}'_k$ ($k \neq i, j$), $\vec{A}'_i = \vec{A}_i \cos\alpha + \vec{A}_j \sin\alpha$ and $\vec{A}'_j = \vec{A}_j \cos\alpha - \vec{A}_i \sin\alpha$.

To satisfy our second criterion, we do not choose rotation dimensions among the dimensions of pre-trained word embeddings. Rather, to prevent the deterioration of already improved relations, we use the principle of family action: *if a transformation is applied for one ball, the same transformation will be applied to all its child balls*. Among the three transformations, only **H-tran** preserves inclusion and disconnectedness relations among the family of $\mathcal{N}$-balls as illustrated in Figure 3(d), therefore **H-tran** has the priority to be used.

In the process of adjusting sibling $\mathcal{N}$-balls to be disconnected from each other, we apply **H-tran** obeying the principle of family action. When an $\mathcal{N}$-ball is too close to the origin of the space, a **S-tran** plus **R-tran** will be applied which may change the pre-trained word embeddings.

---

[2]This phenomenon appears even in very small data sets. We developed a visual simulation for illustration. The source code is available at `https://github.com/GnodIsNait/bp94nball.git`

Following the depth-first procedure, a parent ball is constructed after all its child balls. Given a child $\mathcal{N}$-ball $\mathbb{B}(\vec{B}, r_2)$, a candidate parent ball $\mathbb{B}(\vec{A}, r_1)$ is constructed as the minimal cover of $\mathbb{B}(\vec{B}, r_2)$, illustrated in Figure 3(b). The final parent ball $\mathbb{B}(\vec{P}, r_p)$ is the minimal ball which covers these already constructed candidate parent balls $\mathbb{B}(\vec{P_i}, r_{p_i})$.

---

**Algorithm 1:** $training\_one\_family$(root): Depth first algorithm to construct $\mathcal{N}$-balls of all nodes of a tree

---

**input** : a tree pointed by root; each node stores a word sense and its word embedding
**output**: the $\mathcal{N}$-ball embedding of each node

children $\longleftarrow get\_all\_children\_of$(root)
**if** $number\_of$(children) $> 0$ **then**
    **foreach** child $\in$ children **do**
        `// depth first`
        $training\_one\_family$(child)
    **end**
    **if** $number\_of$(children) $> 1$ **then**
        `// adjusting siblings to be disconnected from each other`
        $adjust\_to\_be\_disconnected$(children)
    **end**
    `// create parent ball for all children`
    root $= create\_parent\_ball\_of$(children)
**else**
    $initialize\_ball$(root)
**end**

---

## 4 EXPERIMENTS AND EVALUATIONS

### 4.1 EXPERIMENT 1: CONSTRUCT $\mathcal{N}$-BALL EMBEDDINGS FOR LARGE-SCALE DATA SETS

We use the GloVe word embeddings of Pennington et al. (2014) as the pre-trained vector embedding of words and extract trees of hyponym relations among word senses from Word-Net 3.0 of Miller (1995). The data set has $54,310$ word senses and 291 trees, among which root *entity.n.01* is the largest tree with $43,669$ word senses. Source code and input data sets are publically available at `https://github.com/GnodIsNait/nball4tree.git`. We proved that all subordinate relations in the category tree are preserved in $\mathcal{N}$-ball embeddings. That is, zero energy cost is achieved by utilizing the proposed geometric approach. Therefore, the first criterion is satisfied.

### 4.2 EXPERIMENT 2: TEST IF PRE-TRAINED WORD-EMBEDDINGS ARE WELL PRESERVED

We apply homothetic, shifting, and rotating transformations in the construction/adjusting process of $\mathcal{N}$-ball embeddings. Shifting and rotating transformations may change pre-trained word embeddings. The aim of this experiment is to examine the effect of the geometric process on pre-trained word embeddings, and check whether the second criterion is satisfied.

**Method 1** We examine the standard deviation (std) of the pre-trained word embedding in $\mathcal{N}$-ball embeddings of its word senses. The less their std, the better they are preserved.

The $\mathcal{N}$-ball embeddings have 32,503 word stems. For each word stem, we extract the word embedding parts from $\mathcal{N}$-ball embeddings, normalize them, minus pre-train word embeddings, and compute standard deviation.

The maximum std is 0.7666. There are 417 stds greater than 0.2, 6 stds in the range of $(0.1, 0.2]$, 9 stds in the range of $(10^{-12}, 0.1]$, 9699 stds in the range of $(0, 10^{-12}]$, 22,372 stds equals 0. With this statistics we conclude that only a tiny fraction (1.3%) of pre-trained word embeddings have a small change (std $\in (0.1, 0.7666]$).

**Method 2** The quality of word embeddings can be evaluated by computing the consistency (Spearman's co-relation) of similarities between human-judged word relations and vector-based word similarity relations. The standard data sets in the literature are the WordSim353 data set (Finkelstein et al., 2001) which consists of 353 pairs of words, each pair associated with a human-judged value about the co-relation between two words, and Stanford's Contextual Word Similarities (SCWS) data set (Huang et al., 2012) which contains 2003 word pairs, each with 10 human judgments on the similarity. Given a word $w$, we extract the word embedding part from its word senses' $\mathcal{N}$-ball embeddings and use the average value as the word-embedding of $w$ in the experiment.

Unfortunately, both data sets cannot be used directly within our experimental setting as some words do not appear in the ball-embedding due to (1) words whose word senses have neither hypernym, nor hyponym in Word-Net 3.0, e.g. *holy*; (2) words whose word senses have different word stems, e.g. *laboratory, midday, graveyard, percent, zoo, FBI, . . .* ; (3) words which have no word senses, e.g. *Maradona*; (4) words whose word senses use their basic form as word stems, e.g. *clothes, troops, earning, fighting, children*. After removing all the missing words, we have 318 paired words from WordSim353 and 1719 pairs from SCWS dataset for the evaluation.

We get exactly the same Spearman's co-relation values in all 11 testing cases: the Spearman's co-relation on WordSim318 is 76.08%; each test on Spearman's co-relations using SCWS1719 is also the same. We conclude that $\mathcal{N}$-ball embeddings are a "loyal" extension to word embeddings, therefore, the second criterion is satisfied.

### 4.3 Experiment 3: Qualitative Evaluation of $\mathcal{N}$-ball Embeddings

Following Levy & Goldberg (2014), we do qualitative evaluations. We manually inspect nearest neighbors and compare results with pre-trained GloVe embeddings. A sample is listed in Table 1 - 2 with interesting observations as follows.

**Precise neighborhoods** $\mathcal{N}$-ball embeddings precisely separate word senses of a polysemy. For example, the nearest neighbors of berlin.n.01 are all cities, the nearest neighbors of berlin.n.02 are all names as listed in Table 1.

**Typed cosine similarity function better than the normal cosine function** $Sim_0$ enriched by category information produces much better neighborhood word senses than the normal cosine measurement. For example, the top-5 nearest neighbors of *beijing* in GloVe using normal cosine measurement are: *china*, *taiwan*, *seoul, taipei,* *chinese*, among which only *seoul* and *taipei* are cities. The top-10 nearest neighbors of *berlin* in GloVe using normal cosine measurement are: *vienna, warsaw, munich, prague,* *germany*, *moscow, hamburg, bonn, copenhagen, cologne*, among which *germany* is a country. A worse problem is that neighbors of the word sense berlin.n.02 as the family name do not appear. Without structural constraints, word embeddings are severely biased by a training corpus.

**Category information contributes to the sparse data problem** Due to sparse data, some words with similar meanings have negative cosine similarity value. For example, *tiger* as a fierce or audacious person (tiger.n.01) and *linguist* as a specialist in linguistics (linguist.n.02) seldom appear in the same context, leading to $-0.1$ cosine similarity value using GloVe word embeddings. However, they are hyponyms of person.n.01, using this constraint our geometrical process transforms the $\mathcal{N}$-balls of tiger.n.01 and linguist.n.02 inside the $\mathcal{N}$-ball of person.n.01, leading to high similarity value measured by the typed cosine function.

**Upper category identification** Using $\mathcal{N}$-ball embeddings, we can find upper-categories of a word sense. Given word sense $ws$, we collect all those $cat$s satisfying $\mathcal{D}_{inside}(ws, cat) > 0$. These $cat$s shall be upper-categories of $ws$. If we sort them in increasing order and mark the first $cat$ with $+_1$, the second with $+_2 \ldots$, the $cat$ marked with $+_1$ is the direct upper-category of $ws$, the $cat$ marked with $+_2$ is the direct upper-category of the $cat$ with $+_1 \ldots$, as listed in Table 2.

| word sense 1 | word sense 2 ($Sim_0$, cos) |
|---|---|
| beijing.n.01 | london.n.01 ($>$0.99, 0.47), atlanta.n.01 ($>$0.99, 0.27) washington.n.01 ($>$0.99, **-0.11**), paris.n.0 ($>$0.99, 0.46), potomac.n.02 ($>$0.99, 0.18), boston.n.01($>$0.99, 028) |
| berlin.n.01 | madrid.n.01 ($>$0.99, 0.47), toronto.n.01 ($>$0.99, 0.46), rome.n.01 ($>$0.99, 0.68), columbia.n.03 ($>$0.99, 0.39), sydney.n.01 ($>$0.99, 0.52), dallas.n.01($>$0.99, 0.28) |
| berlin.n.02 | simon.n.02 ($>$0.99, 0.34), williams.n.01 ($>$0.99, 0.24), foster.n.01, ($>$0.99, 0.13), dylan.n.01 ($>$0.99, 0.10), mccartney.n.01 ($>$0.99, 0.23), lennon.n.01($>$0.99, 0.25) |
| tiger.n.01 | survivor.n.02 ($>$0.99, 0.40), neighbor.n.01 ($>$0.99, 0.36), immune.n.01($>$0.99, 0.10), linguist.n.02 ($>$0.99, **-0.1**), bilingual.n.01 ($>$0.99, **-0.06**), warrior.n.01 ($>$0.99, 0.68) |
| france.n.02 | white.n.07($>$0.99, 0.31), woollcott.n.01($>$0.99, **-0.12**), uhland.n.01($>$0.99, **-0.32**), london.n.02($>$0.99, 0.52), journalist.n.01($>$0.99, 0.33), poet.n.01($>$0.99, 0.20) |
| cat.n.01 | tiger.n.02($>$0.99, 0.62), fox.n.01($>$0.99, 0.44), wolf.n.01($>$0.99, 0.67), wildcat.n.03($>$0.99, 0.16), tigress.n.01($>$0.99, 0.40), vixen.n.02($>$0.99, 0.38) |
| y.n.02 | q.n.01($>$0.99, 0.45), delta.n.03($>$0.99,0.33), n.n.05($>$0.99, 0.60), p.n.02($>$0.99, 0.44), f.n.04($>$0.99, 0.55), g.n.09($>$0.99, 0.52) |

Table 1: Top-6 nearest neighbors based on $Sim_0$, the cos value of word stems are listed, e.g. cos(beijing, london)$= 0.47$, tiger.n.01 refers to a fierce or audacious person.

| word sense 1 | word sense 2 |
|---|---|
| beijing.n.01 | city.n.01$+_1$, municipality.n.01$+_2$, region.n.03$+_3$, location.n.01$+_4$, object.n.01$+_5$, entity.n.01$+_6$ |
| berlin.n.02 | songwriter.n.01$+_1$, composer.n.01$+_2$, musician.n.02$+_3$, artist.n.01$+_4$, creator.n.02$+_5$ |
| tiger.n.01 | person.n.01$+_1$, organism.n.01$+_2$, whole.n.02$+_3$, object.n.01$+_4$, entity.n.01$+_5$ |
| france.n.02 | writer.n.01$+_1$, communicator.n.01$+_2$, person.n.01$+_3$, organism.n.01$+_4$ |
| cat.n.01 | wildcat.n.03$+_1$, lynx.n.02$+_2$, cougar.n.01$+_3$, bobcat.n.01$+_4$, caracal.n.01$+_5$, ocelot.n.01$+_6$, feline.n.01$+_7$,jaguarundi.n.01$+_8$ |
| y.n.02 | letter.n.02$+_1$ character.n.08$+_2$ symbol.n.01$+_3$ , signal.n.01$+_4$ communication.n.02$+_5$ |

Table 2: $+_k$ represents the $k^{th}$ minimal positive value of $\mathcal{D}_{inside}$.

## 4.4 Experiment 4: Membership Validation

The fourth experiment is to validate the category of an unknown word, with the aim to demonstrate the predictive power of the embedding approach (Baroni et al., 2014). We describe the task as follows: Given pre-trained word embeddings $\mathcal{E}_N$ with vocabulary size $N$, and a tree structure of hyponym relations $\mathcal{T}_K$ on vocabulary $W_K$, $(K < N)$. Given $w_x \notin W_K$ and $c \in W_K$, we need to decide whether $w_x \in c$. For example, when we read *mwanza is also experiencing*, we may guess *mwanza* a person, if we continue to read *mwanza is also experiencing major infrastructural development*, we would say *mwanza* is a city. If *mwanza* is not in the current taxonomy of hypernym structure, we only have its word embedding from the text. Should *mwanza* be a city, or a person?

**Dataset**  From 54,310 word senses, we randomly selected 1,000 word senses of nouns and verbs as target categories, with the condition that each of them has at least 10 direct hyponyms; For each target category, we randomly select $p\%$ ($p \in [5, 10, 20, 30, 40, 50, 60, 70, 80, 90]$) from its direct hyponyms as training data. The test data sets are generated from three sources: (1) the rest $1 - p\%$ from the direct hyponyms as true values, (2) randomly choose 1,000 false values from $W_K$; (3) 1,000 words from $W_N$ which do not exist in $W_K$. In total, we created 118,938 hyponymy relations in the training set, and 17,975,042 hyponymy relations in the testing set.

**Method**  We develop an $\mathcal{N}$-ball solution to solve the membership validation task as follows: Suppose $c$ has a hypernym path $[c, h_1, h_2, \ldots, h_m]$ and has several known members (direct hyponyms) $t_1, \ldots, t_s$. For example, *city.n.01* has a hypernym path [*city.n.01, municipality.n.01, urban_area.n.01, …, entity.n.01*] and a number of known members *oxford.n.01, banff.n.01, chicago.n.01*. We construct $\mathcal{N}$-ball embeddings for this small tree with the stem $[c, h_1, h_2, \ldots, h_m]$ and leaves $t_1, \ldots, t_s$, and record the history of the geometric transformations of each ball. Suppose that $w_x$ be a member of $c$, we initialize the $\mathcal{N}$-ball of $w_x$ using the same parameter as the $\mathcal{N}$-ball of $t_1$, and apply the recorded history of the geometric transformations of $t_1$'s $\mathcal{N}$-ball for $w_x$'s $\mathcal{N}$-ball. If the final $\mathcal{N}$-ball of $w_x$ is located inside the $\mathcal{N}$-ball of $c$, we will decide that $w_x$ is the member of $c$, otherwise not. This method can be explained in terms of Support Vector Machines (Shawe-

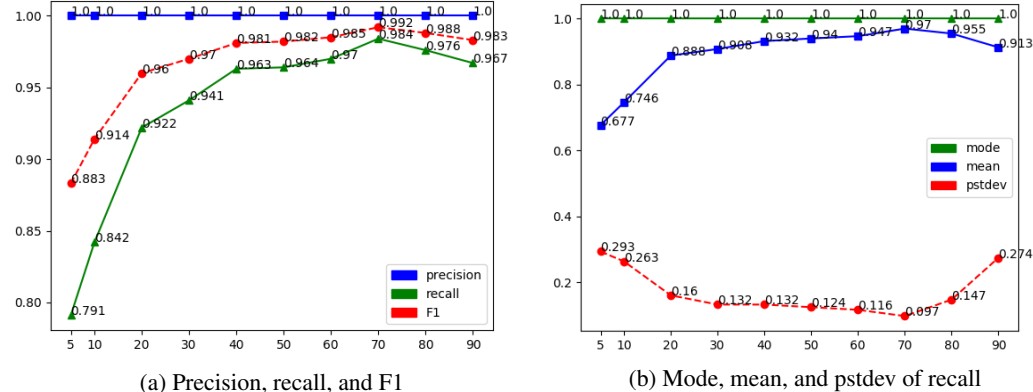

Figure 4: (a) Precision, recall, F1 score of hypernym prediction, when $5\%, 10\%, 20\%, \ldots, 90\%$ of the members are used for training; (b) Mode, mean, and population standard deviation of the recall, when $5\%, 10\%, 20\%, \ldots, 90\%$ of the members are used for training.

Taylor & Cristianini, 2004) as follows: the boundary of $c$'s $\mathcal{N}$-ball is supported by all $\mathcal{N}$-balls of its known members. If the unknown word were a member of $c$, its $\mathcal{N}$-ball shall have the same PLC as the member of $c$, and according to the principle of family action, it shall have the same geometric transformation, and will contribute one candidate parent ball. We can introduce a ratio $\gamma \geq 1$ to zoom-out the boundary of $c$'s $\mathcal{N}$-ball.

**Evaluation and Analysis** In the experiment, results show that the $\mathcal{N}$-ball method is very precise and robust as shown in Figure 4(a): the precision is always 100%, even only 5% from all members is selected as training set. The method is quite robust: If we select 5% as training set, the recall reaches 76.8%; if we select 50% as training set, the recall reaches 96.7%. Theoretically, the $\mathcal{N}$-ball method can not guarantee 100% recall as shown in Figure 4(b). If $p < 70\%$, the population standard deviation (pstdev) decreases with the increasing percentage of selected training data. When $p > 70\%$, there is a slight increase of pstdev. The reason is that in the experiment setting, if more than 80% of the children are selected, it can happen that only one unknown member is left for validating. If this single member is excluded outside of the category's $\mathcal{N}$-ball, the recall drops to 0, which increases pstdev. The experiment result can be downloaded at `https://figshare.com/articles/membership_validation_results/7571297`.

In the literature of representational learning, margin-based score functions are the state-of-the-art approach (Gutmann & Hyvärinen, 2012): The score of a positive sample shall be larger than the score of a negative sample plus a margin. This can be understood as a simple use of categorization – no chained subordinate relations, no clear membership relations of negative samples, no requirement on zero energy loss. However, when category information is fully and strictly used, the precision will increase significantly, and surprisingly reach 100% in this experiment.

## 5 RELATED WORK

Lenci & Benotto (2012) explored the possibility of identifying hypernyms in distributional semantic model; Santus et al. (2014) presented an entropy-based model to identify hypernyms in an unsupervised manner; Kruszewski et al. (2015) induced mappings from words/sentences embeddings into Boolean structure, with the aim to narrow the gap between co-occurrence based embeddings and logic-based structures. There are some works on word embedding and knowledge graph embedding using regions to represent words or entities. Athiwaratkun & Wilson (2017) used multi-modal Gaussian distribution to represent words; He et al. (2015) embedded entities using Gaussian distributions; Xiao et al. (2016) used manifolds to represent entities and relations; Nickel & Kiela (2017) used Poincaré balls to embed tree structures. Mirzazadeh et al. (2015) share certain common interest with the presented work in embedding constraints. However, in none of these works, structural imposition at zero-energy cost is targeted.

# 6 CONCLUSION AND ON-GOING WORK

We proposed a novel geometric method to precisely impose external tree-structured category information onto word embeddings, resulting in region-based ($\mathcal{N}$-ball embeddings) word sense embeddings. They can be viewed as Venn diagrams (Venn, 1880) of the tree structure, if zero energy cost is achieved. Our $\mathcal{N}$-ball method has demonstrated great performance in validating the category of unknown words, the reason for this being under further investigation. Our on-going work also includes multi-lingual region-based knowledge graph embedding where multiple relations on directed acyclic graphs need to be considered. $\mathcal{N}$-balls carry both vector information from deep learning and region information from symbolic structures. Therefore, $\mathcal{N}$-balls establish a harmony between Connectionism and Symbolicism, as discussed by Marcus (2003), and thus may serve as a novel building block for the commonsense representation and reasoning in Artificial Intelligence. $\mathcal{N}$-balls, in particular, contribute a new topic to Qualitative Spatial Reasoning (QSR) that dates back to Whitehead (1929).

## ACKNOWLEDGMENTS

Partial financial support from NSFC under Grant No. 61472177 and 61661146007, and from BMBF (Bundesministerium für Bildung und Forschung) of Germany under grant number 01/S17064 and 01/S18038C are greatly acknowledged. We are greatly indebted to Dagstuhl Seminar 15201 *Cross-LingualCross-Media Content Linking: Annotations and Joint Representations* for fruitful and interesting discussions. We are also thankful for critical comments from three anonymous reviewers, and for the OpenReview policy.

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
