# OpenReview forum: "Imposing Category Trees Onto Word-Embeddings Using A Geometric Construction"
_ICLR.cc/2019/Conference_

### Official Review · AnonReviewer3 · 2018-11-02
**Proposal of N-ball embedding for tree structures**

**Rating:** 3
**Confidence:** 4

**Review:**

This paper proposes N-ball embedding for taxonomic data. An N-ball is a pair of a centroid vector and the radius from the center, which represents a word.

Major comments:

- The weakness of this paper is lack of experimental comparisons with other prominent studies. The Poincare embedding and the Lorentz model are recently proposed and show a good predictive performance in hypernymy embedding.
- WordNet concepts are actually structed in DAG. Recent studies on structure embedding can hadle DAG data. It is not clear how to extend N-ball embedding for handling DAT structures.

- Related work is not sufficiently described.

- It is not clear why N-ball embedding is suitable for hierarchical structures.

---

### Official Review · AnonReviewer1 · 2018-11-02
**Nice geometrical observations, but not novel and have insufficient empirical evaluation of the quality**

**Rating:** 4
**Confidence:** 5

**Review:**

Attention!!! This submission contains Github and Google Drive links to author-related accounts (see e.g. the abstract). I do not think this is permitted or standard. I leave the decision regarding "automatic rejection" of the submission to meta-reviewers of the paper.
------------------------------------------------
The paper presents a method for tweaking existing vector embeddings of categorical objects (such as words), to convert them to ball embeddings that follow hierarchies. Each category is represented as a Eucldiean norm ball in high dimensional space, with center and radii adaptable to data. Next, inclusion and exclusion constraints on each pair of  balls are imposed based on the hierarchical structure. These constraints are imposed via an algorithmic approach.  The empirical study includes investigating the consistency of the representation with the hierarchy and demonstrating nearest neighbors for a set of words.

On the positive side, the paper addresses an important problem. It is readable and well organized. The related work could be improved by adding a number of representative related works such as [3,4].

The major concern about the paper is the originality of the method. Encoding hierarchies with high dimensional balls and encoding inclusion and exclusion as constraints on those balls is a neat and powerful idea from modeling perspective. However, it is not novel, since the approach is already established for example in [1, 2 Chapter 5].
The next major concern is regarding the evaluation of the quality of embeddings.
The empirical evaluation does not sufficiently evaluate the quality of tweaked embeddings. In contrast, the quantitative evaluation is more concerned with if the embeddings being consistent with the given hierarchy. In particular, not enough quantitative evidence that the proposed embeddings are actually effective in capturing semantics or in prediction tasks is provided. It should be noted that, the first aspect, ie consistency of the feasible solutions with hierarchy, can be theoretically established (see e.g. [1]).  The first paragraph of 3.2 seems unclear or wrong. See for example [2] for a gradient based solution for the problem.
Finally, using an algorithmic approach as opposed to learning method for constructing embeddings, makes the method not directly related to the topic of ICLR conference.

Overall, due to the above reasons, I vote the paper to be rejected. (The poor anonymization makes it a strong case for a reject.)

[1] Mirzazadeh, F., Ravanbakhsh S., Ding N., Schuurmans D.,  "Embedding inference for structured multilabel prediction", NIPS 2015.
[2] Mirzazadeh, F."Solving Association Problems with Convex Co-embedding", PhD thesis, 2017. (Chapter 5)
[3] Vilnis, Luke, and Andrew McCallum. "Word representations via gaussian embedding.", ICLR 2015.
[4] Vendrov, I., Kiros, R., Fidler, S., Urtasun, R. "Order-embeddings of images and language." ICLR 2016.

---

> ### Author Response · Authors · 2018-11-21
> **the anonymization is carefully taken cared of**
>
> thanks for the detailed comments, though not acceptable.
>
> the anonymization issue is carefully taken cared of.  the github account is temporarily created for this projects. readers/reviews cannot figure out who we are.
>
> on the other hand, the iclr submission policy allows papers that have appeared on non-peered reviewed websites (like arXiv).  therefore, anonymization  is not an issue.

---

> > ### Public Comment · (anonymous) · 2018-12-22
> > **not really...**
> >
> > The github page has commits by one of the authors, and the google drive is owned by the same person.

---

> > > ### Author Response · Authors · 2018-12-23
> > > **Re: not really...**
> > >
> > > do not see the line of your argument.

---

> > > > ### Public Comment · (anonymous) · 2019-01-07
> > > > **AC/PC Please Note Lack of Anonymity**
> > > >
> > > > @AC/PC This paper flouts ICLR's anonymity guidelines. As the comment by the review, and the other anonymous poster says, the GitHub page lists the first author's name under "contributors", and the Google Drive lists the first author's name as well. It's likely a careless error, and may not warrant rejection, but AC/PC should discuss this with the author, and comment on this in the meta-review as a warning.
> > > >
> > > > The author's responds to this is poor e.g. "therefore, anonymization  is not an issue.", "do not see the line of your argument. ". Especially since the authors did not issue an apology, this should be flagged as a very important point.

---

> > > > > ### Author Response · Authors · 2019-01-07
> > > > > **two points have been neglected by the reviewer**
> > > > >
> > > > > (1) this year's submission policy accepts papers from arXiv. "papers that have appeared on non-peered reviewed websites (like arXiv) or that have been presented at workshops (i.e., venues that do not have a publication proceedings) do not violate the policy". So, you can detect authors of submissions that exist in arXiv.
> > > > >
> > > > > (2) the detective work of your kind may round up names with less or no claims to authorship, because they may be master students from our institute or partner universities who help data collection and implementation.

---

> > > > > > ### Public Comment · (anonymous) · 2019-01-22
> > > > > > **Master students' deserve no crediting ?!**
> > > > > >
> > > > > > So let me get this right. It is perfectly normal to not include people who are directly involved with curating/creating the data and carrying out part of the implementation ? which themselves are two very core tasks in machine learning.
> > > > > >
> > > > > > I think in most modern institutions it is perfectly normal to expect the names found in the official GitHub repository to be names of authors directly involved in the paper.
> > > > > >
> > > > > > It seems unlikely to find all commits in a work pointing to a single name (which is not of a masters student but of a senior research fellow).
> > > > > >
> > > > > > Such detective work should be a valid point to raise, and it was not that thorough specially since the anonymous account is the name of the first author in reverse. You definitely don't need to be Sherlock Holme's to spot that.

---

### Official Review · AnonReviewer2 · 2018-11-02
**Interesting task but weak evaluation mainly dominated by qualitative analysis**

**Rating:** 4
**Confidence:** 4

**Review:**

This paper focuses on adjusting the pretrained word embeddings so that they respect the hypernymy/hyponymy relationship by appropriate n-ball encapsulation. They propose to do so by augmenting the word embeddings with information from a resource like Wordnet and applying 3 kinds of geometric transformations to enforce the encapsulation.

The motivation of doing this is not very clear and experimental results are mainly qualitative (and subjective) showing that hypernymy relation can be predicted and preserved by their adjustment. Since, this work relies on Wordnet, the coverage of vocabulary is severely limited and as the authors discus in the results with the section titled "Experiment 3: Method 2", they had to remove many words in the standard semantic similarity datasets which casts shadow on the usefulness of the proposed approach. It is unclear what the main contribution of such an approach.

Apart from this, the paper is diffcult to read and some parts (especially those pertaining to Figure 3) encode a simple concept that has been expressed in a very complicated manner.

Overall, I give a score of 4 because of the limited coverage of the approach because of reliance on Wordnet and inadequate empirical evidence of usefulness of this approach.

---

### Author Response · Authors · 2018-11-11
**the main point of the work is unfortunately neglected by all reviewers**

Thanks for the comments. Unfortunately, three reviewers misunderstood or neglected the main point of this work.

Our work is targeted at the possibility of strictly imposing structures into  deep-learning systems. Such imposing must achieve global minimum (zero energy cost).  We imposed external tree structures into word-vectors produced by neural networks, achieving zero energy cost (this has not been achieved in the literature of representational learning). Our embedding also leads to perfect experimental results – no deep-learning systems in the literature has achieved 100% precision, as their training processes are only targeted at local minimums.  We would not use models, which only achieving local minimum, to compare with our model. This is not our aim.

Our aim is to perfectly impose external structures into vectors produced by deep-learning models, while keeps these vectors unchanged as much as possible, which can be reviewed as an optimization for representation learning (one of the topics of iclr-19).

---

> ### Public Comment · (anonymous) · 2019-01-22
> **100% precision and low recall?**
>
> Any model can have high precision with low recall.
>
> Then what criteria to categorize models?
>
> A scientist with integrity should refer to the base theory (Venn's diagram, set theories) *and*  constructively compare to previous work instead of trying to avoid.

---

### Public Comment · (anonymous) · 2019-01-21
**Name reversal is not a valid anonymization technique**

Apart from the multitude of issues that the reviewers have raised with this work, it seems that the authors indeed broke the double-blind review process. The link https://github.com/gnodisnait/bp94nball.git in the paper abstract clearly contains the username gnodisnait, which is the name of the main author Tiansi Dong spelled in reverse. I urge the area chair to reconsider their acceptance decision based on this very obvious and intentional act of breaking ICLR rules.

---

> ### Author Response · Authors · 2019-01-22
> **this issued has been replied blow**
>
> here is the earlier reply:
> (1) this year's submission policy accepts papers from arXiv. "papers that have appeared on non-peered reviewed websites (like arXiv) or that have been presented at workshops (i.e., venues that do not have a publication proceedings) do not violate the policy". So, you can detect authors of submissions that exist in arXiv.
>
> (2) the detective work of your kind may round up names with less or no claims to authorship, because they may be master students from our institute or partner universities who help data collection and implementation.
>
> ------
> according to the policy, it is possible for reviewers to identify authors. in this case, you raised the reverse technique, after the real names are published, instead of the reviewing process. That is, the anonymization technique actually works.

---

> > ### Public Comment · (anonymous) · 2019-01-22
> > **RE: this issued has been replied below**
> >
> > blow ?
> >
> > Anyways. Whilst non proceeding publications are allowed, that  does not override the anonymity rule.  Whilst the authors can be googled for, that does not mean that the author should post two links containing their name (just clicking on it leads to the name, and reading it also leads to the name ... ).
> >
> > Submission instructions clearly state:
> >
> > > Submissions and reviews are both anonymous.
> >
> > Having the soft constraint on non-proceeding publications does not override this instruction. Even if you have an arXiv entry you should not be linking it in the papers page, or any other information that makes the submission not be anonymous.
> >
> > Going away and searching for the paper on arXiv is not the same as clicking a link that directly reveals the authors name. The attempt to follow the conferences rules seems deliberately poorly executed by the author.
> >
> > Has ICLR reached the point where its rules can be completely disregarded, without comment ? This issue should be discussed and addressed by the PC's , if its ok to break such rules then the should be removed or adapted appropiately.

---

### Public Comment · (anonymous) · 2019-01-22
**Isn't it a joke for AC to accept the paper given all the reviewers vote for rejection?**

Especially when at least one of the anonymous reviewers is an expert in this subfield and gave strong and compelling arguments for rejection.

---

> ### Author Response · Authors · 2019-01-22
> **it is also an issue raised some weeks ago**
>
> the point is that all reviewers neglect the main point of this paper : a novel geometric method to perfectly impose structures onto vector space -- using higher dimensional regions than vectors.

---

### Public Comment · (anonymous) · 2019-01-23
**Can the AC participate in the discussion or at least explain the acceptance decision?**

The AC's comment does not have any technical content, in particular, there is nothing that addresses any of the reviewers concerns, including Rev 1, who points out that the paper fails to cite prior work that already introduces concepts this paper is trying to present as proposed. There is no comment on this discussion page that would defend the authors' claim.

In addition, the authors failed to comply with the anonymization policy. It takes three link clicks to get from the submitted manuscript to see 4 of the authors' names (go to the "mushroom" repo, the names are at the bottom of the page). Authors' comments reveal that they are under the impression that allowing Arxiv preprints invalidates the rest of the anonymity policy, which raises questions of whether they read the author's guidelines presented at the conference website iclr.cc.

The authors and Rev 1 failed to reach a consensus, while the authors were openly belligerent in both discussions that happened. See comments as "your feeling is understandable", "unacceptable comment", and "the detective work of your kind".

--------––------------------------––------------------------––------------------------––------------------------––------------------------––----------------

Given that any single point of the above is substantive grounds for a rejection, the lack of a transparent decision by AC feels out of place here. OpenReview exists in part so that the reviewer entities can be held publicly accountable for their reviews. This is not what's happened here.

---

> ### Comment · Area_Chair1 · 2019-01-24
> **More details**
>
> I feel like my comments already said everything needed.
> But here's more.
>
> The work is very important and has potential to bridge the gap between neural and logical reasoning methods. In my eyes that could have major impact on the field. I see my job as area chair not in just defining a cutoff based on scores. I've worked in neural nlp for a while and happen to disagree with the reviewers here.
> It's the only paper in my batch for which I came to this conclusion.
> I do believe the content matters more than the formalities.
>
> Most reject reasons here are not even about the actual content and method of the paper.
>
> All related work has been or will be added to a later paper version so having missed it in prior submission is now irrelevant. That's what this process is for.
> If the authors still did not cite relevant work it would be bad.
>
> Searching for a title of a paper on Google is much easier than clicking through github repos and the reverse name github thing is only obvious in retrospect. So it's not relevant in my opinion.
>
> Since people speculate on Twitter:
> I have never heard of the authors and don't know any of them personally.
>
> I understand that people get upset if their papers get rejected or related work is not cited often or well enough. I've had many papers rejected in the past too. I've also seen papers related to my papers that didn't cite me properly. It happens. Most of the time it's an honest mistake that is easily fixable.
>
> That being said I'm surprised people spend so much time and energy trying to reject a paper...
> I hope we could spend this energy on novel research instead.

---

> > ### Public Comment · (anonymous) · 2019-01-24
> > **Re: More details**
> >
> > Three reviewers took an effort to provide feedback. Some reviewers were more elaborate than others but the authors didn't provide a direct response to any of the reviewers (as far as we can see publicly), except regarding the anonymization issues (starting off with "thanks for the detailed comments, though not acceptable.").
> > None of the content-related comments of reviewers seem to have been addressed by the authors publicly.
> > Instead, one week after the reviews, the authors claim that all three reviewers neglected the main point of the paper, in a top-level comment.
> >
> > Only one reviewer raised the anonymity issues on the linked resources. Whether or not knowledge of author identity would bias reviewers seems less of a point than the fact that it violates submission rules, which others do have to obey. Another issue of the same reviewer is similarity to previous work, which was not discussed until after the acceptance decision. Other issues about evaluation and comparison to previous work sound valid but have not been further addressed by the authors.
> >
> > The acceptance decision would've been understandable if the authors at least tried to convince the reviewers by addressing the issues they have about their content (which didn't happen as far as we can see publicly), and if the final review scores weren't at low as 4, 4, 3.
> > From how it looks publicly, it seems the reviewers' efforts were simply ignored.
> >
> > That being said, the time the reviewers used to review this work could've been used for novel research instead. And maybe this gets so much attention elsewhere because people are genuinely surprised that this actually happened at such a top venue as ICLR.

---

> > > ### Comment · Area_Chair1 · 2019-01-24
> > > **Reviews are important**
> > >
> > > The reviewers raised important points.
> > > Detailed equations were discussed and improved our understanding of the paper.
> > > Please read the below discussions that are public.
> > >
> > > I'm not saying this should get a best paper award. I'm not saying this should be an oral.
> > >
> > > But in my opinion, despite its flaws, and having worked and thought about this field for over a decade, it is my opinion that this paper could be more important than many others.
> > > It may present a puzzle piece to solve one of the major issues of AI. Connecting neural and logical/set reasoning will be impactful and may help with common sense reasoning.
> > >
> > > It is not the most perfect piece of research but outlines an interesting direction and I think will be beneficial to be discussed at the conference.
> > >
> > > Especially to also draw attention to the so far barely cited papers that are being discussed in these reviews.
> > > These good papers too will be lifted and get more exposure.
> > >
> > > I can tell you that everybody here was listened to. Nobody was ignored.
> > > I believe it is a sign of quality that ACs do not simply use a threshold to make accept/reject decisions but try their best to think about impact.
> > >
> > > We do not want to be the kind of field or conference that hold back impactful ideas and directions because they have issues.
> > >
> > > I do encourage all parties to stay professional and friendly. Even when  anonymous.

---

> > > > ### Public Comment · (anonymous) · 2019-01-24
> > > > **Reviews are important ?**
> > > >
> > > > Repeating "It may present a puzzle piece to solve one of the major issues of AI." again and again does not sound like an objective comment/review, specially compared to the extensive and quantitive reviews given by the anonymous reviewers. Looking at the AC's comments I can't really gain an intuition as to why the paper was accepted, none of the reviewers concerns are addressed, no discussion attempt was made with the reviewers in the portal. How does this reflect reviews are important ?
> > > >
> > > > Furthermore sharing links containing the authors name is in clear violation of the anonymity rule. Interacting with the GitHub repository is definitely something natural to do for the author, have a quick check of how reproducible the work is, maybe how much work was put into making it reproducible. Some conferences like ICML encourage this heavily. Thus the statement made by the AC :
> > > >
> > > > "Searching for a title of a paper on Google is much easier than clicking through github repos and the reverse name github thing is only obvious in retrospect. So it's not relevant in my opinion."
> > > >
> > > > Is heavily overlooking the conferences rules.  Why does everyone else have to make a proper effort in doing this ? and yet we have an exception for this paper.
> > > >
> > > > "Anonymity. ICLR is double-blind, which means that authors are not aware of reviewer identities and reviewers are not aware of author identities. If you believe a paper contains an anonymity violation, contact your AC immediately. Anonymity violations are not considered as part of reviewing criteria, they are requirements for submission. Unless your AC decides that the paper does indeed violate anonymity, proceed to review it as normal. Do not reveal your or the authors’ identities in the discussion."
> > > >
> > > > Furthermore:
> > > >
> > > > "Arxiv and prior work. ICLR considers unpublished arxiv papers to be prior work. While we encourage reviewers to apply the reasonable standards of the relevant community in considering what does and does constitute prior work, the following minimum standards will be enforced: no paper will be considered prior work if it appeared on arxiv, or another online venue, less than 30 days prior to the ICLR deadline."
> > > >
> > > >  The latter exception with Arxiv and non proceedings does not override the former rule regarding anonymous submissions. Furthermore the rule itself has timing constraints, uploading linked prior work to arXiv or GitHub the day before the deadline would be in violation of this rule.

---

> > > > ### Public Comment · (anonymous) · 2019-01-24
> > > > **Lack of technical discussion by AC**
> > > >
> > > > (the author of this comment is the same as the original comment "Can the AC participate in the discussion..." Identities of anonymous authors of other responses in this thread are unknown to me)
> > > >
> > > > My original comment was mainly directed at engaging the AC in the technical discussion. I think it is necessary now to step back and look at the technical content of the paper so that we can "stay professional and friendly". Appeals to the AC's authority and his intuition about the possible impact of the paper might be a valid way of defending the paper, however, since not all of us "have worked and thought about this field for over a decade", it might be hard for some of us to follow the AC's argument. While the AC is completely within his right to accept the paper, my argument is simply that it should be made transparent how the specific technical contribution of the paper affected this decision.
> > > >
> > > > It is understandable that the meta-reviewer doesn't have time to review the paper and has to base the decision on the reviews and the authors comments. If the decision was based on a personal review by the AC, I believe such review (with a summary of technical part and experiments) should be posted so that the decision becomes transparent. It the next part of my comment, I will assume that the decision was instead based on the discussion here, and address AC's comments about this discussion.
> > > >
> > > > --------––------------------------––------------------------––------------------------––------------------------––----------------------
> > > >
> > > > The AC comments that "Detailed equations were discussed and improved our understanding of the paper." While it's true that "equations" were discussed between Rev1 and the authors,  I do not believe our understanding of the paper was improved.
> > > >
> > > > The only publicly visible discussion happened after the acceptance decision. The authors did not elect to address technical concerns of the reviewers before the AC decision, therefore the discussion the AC mentions could not affect the AC's decision. The AC claims that "Nobody was ignored", however, the reviewer comments were ignored by the authors (as far as I can tell from the public part of the discussion).
> > > >
> > > > There was indeed a discussion between Rev1 and authors after the acceptance decision. if you follow the discussion, you will find out that in none of the comments Rev1 agrees with the authors, or the authors agree with any point raised by Rev1. The authors failed to defend their point, and withdrew from the discussion (the last comment is by Rev1). I, therefore, cannot follow the argument of AC that "Detailed equations were discussed and improved our understanding of the paper."
> > > >
> > > > The AC is under the impression that "All related work has been or will be added to a later paper version so having missed it in prior submission is now irrelevant. That's what this process is for.
> > > > If the authors still did not cite relevant work it would be bad." Unfortunately, we have to accept that what is happening is, in fact, "bad". If you follow the discussion, the authors refuse to cite the recent relevant papers, instead stating that they will cite Venn, 1880 and Whitehead, 1929.  They add that "If space available, we consider add one of the two references you mentions which appear in 2015 and 2017", however, the submitted manuscript has not been revised once since submission.
> > > >
> > > > The AC might reasonably state that the fact that the consensus was not reached does not imply rejection of the paper. However, the ICLR AC guidelines state that "A key part of your role during the review period is to actively facilitate discussion among reviewers, with the aim of clarifying aspects of the papers’ claims, and other questions that have been raised, in fair and respectful ways." It seems to me that it would be appropriate for the AC to moderate the discussion between Rev1 and the authors so that the truth can be found. If the AC chooses not to do so, the guidelines still instruct them to " Discuss any major points of contention. As raised by the authors or reviewers in the discussion, and how these might have influenced the decision." None of the AC's comments address or state what the reviewers concerns are and why they have been downweighted in the acceptance decision. I hope that AC chooses to engage in technical discussion (possibly with Rev1 or even the authors) to clarify this.

---

### Public Comment · ~suomynona_noitpurroc1 · 2019-01-24
**Respect ?**

It is incredibly disappointing that this happens in a top tier conference like ICLR. It is also very disrespectful towards all the other authors and reviewers that the rules of the game are completely ignored for this particular paper.

- The main argument of the AC is that he/she "knows" much better than all the reviewers how good this paper is. We all know
 that these reviews are biased, but ignoring 3 reviews for another one it not gonna make things better. All the official reviewers said "reject", but then the AC God came and saved the day, of course, ignoring all the reviewers work.  My question is: why didn't the AC review all the other papers himself, why does he/she need reviewers when it is now clear that the AC is the only person to correctly evaluate the quality of a paper?

- Even if, hypothetically speaking, the AC would hold the absolute truth, this decision is very unfair and disrespectful towards the other submitted papers which were judged based on the 3 reviews, thus subject to the randomness and bias inherent in this process.

- Anonymity break: the github account linked in the abstract could have been easily used to detect the first author's name. This is clearly against the rules and this paper should have been rejected only because of this reason, in the first place.

- Authors outright bullying reviewers: yeah, they have someone to protect them, why care more ?

The ICLR representatives continue up to date to refuse making justice in this case. Moreover, on twitter, they concluded that "we do hope that it is not seen as representative of the 1600 other (mostly) productive conversations that happened concurrently." . I also conclude that: "in life you can succeed by being hard working or by hardly working, but knowing the right people".

---

> ### Public Comment · (anonymous) · 2019-01-25
> **Re: Respect?**
>
> I also do not understand the AC's decision, for all the reasons you mentioned (going against all reviewers clearly rejecting the paper, failure of the authors to formulate rebuttals regarding content issues, seemingly disrespectful attitude of the authors and clear anonymity issues).
>
> However, I don't think mocking the AC or accusing the AC of favoritism is respectful or productive.
>
> I do wish the AC provides clear compelling technical arguments for going against all reviewers and ignoring the other issues.

---

> ### Public Comment · (anonymous) · 2019-01-28
> **Sadness**
>
> The saddest thing to me is that, because of all this drama, this unworthy paper is getting a lot of attention and will likely reach everyone in the community and get cited like crazy in the next few years. On the other hand, other papers, accepted by merit and hard work, are likely to get lost in the archives of the proceedings...

---

> > ### Public Comment · (anonymous) · 2019-01-29
> > **....**
> >
> > ... like tears...  in rain

---

### Meta-Review · Area_Chair1 · 2018-12-13
**Interesting paper on the much studied subject of word vectors**

**Confidence:** 4
**Recommendation:** Accept (Poster)

**Metareview:**

The authors provide an interesting method to infuse hierarchical information into existing word vectors. This could help with a variety of tasks that require both knowledge base information and textual co-occurrence counts.
Despite some of the shortcomings that the reviewers point out, I believe this could be one missing puzzle piece of connecting symbolic information/sets/logic/KBs with neural nets and hence I recommend acceptance of this paper.

---

> ### Comment · AnonReviewer1 · 2018-12-21
> **At least cite the existing work**
>
> The idea is originally established in the works [1, 2]  mentioned in the Review 1 below.
>
> Please check, in particular, Figure 5.1 (in page 62) and  Figure 5.2 (in page 70) of [2] and compare them with Figure 1 and  Figure 2 of the current paper to quickly see the connection.
>
> Hopefully at least citations to the super relevant preceding work is added to the camera ready version of the paper!

---

> > ### Author Response · Authors · 2018-12-21
> > **agree to add reference, original work has the priority**
> >
> > Thanks for the comment.  Your feeling is understandable. We have reviewed Figure 5.1 (in page 62) and Figure 5.2 (in page 70) of [2]. As is described in [2], this idea of representation dates back to Venn digram in 1880, see the caption of Figure 5.1 in [2]. And the region relation as illustrated in Figure 5.2 (in page 70) of [2] dates back the region calculus of Alfred North Whitehead in 1929 in the book „Process and Reality“. We will add this two references in the final version. If space available, we consider add one of the two references you mentions which appear in 2015 and 2017. By the way, neither of the [1] and [2] references John Venn's work in 1880 and Alfred Whitehead's work in 1929.

---

> > > ### Comment · AnonReviewer1 · 2018-12-24
> > > **The need for faithful citations is not about "feelings"**
> > >
> > > Unfortunately there is a misunderstanding. The relevant point of [1, 2]  that needs a faithful citation is not the "reasoning" with "Venn diagrams" or the ancient set theoretic basis.  Rather, the real relevant part is the modern "learning" problem for hierarchies which is already addressed interestingly with a similar method via embeddings. In particular, in [1, 2], each entity is represented with a high dimensional ball with parametrized centers and radii learned from data, as mentioned in the review.
> > >
> > > Notably, it is noticed in [1, 2] that a hierarchy naturally defines inclusion and exclusion constraints on entities of the nodes of the hierarchy tree. Then, these constraints are imposed in a similar vein to your submission by  comparisons of radii and center distances.  The caption you are mentioning contains a mention to an implication of their design that after the embeddings are learned, the learned balls have to comply with the corresponding Venn diagrams corresponding to the parent-child hierarchy in the embedding space for the constraints to actually hold.
> > >
> > > Since the presentations and proofs are algebraic in the [1], seeing the connection with the geometry is harder to general readers, but in [2],  even the geometric argument is highlighted that clarifies the connection.

---

> > > > ### Author Response · Authors · 2018-12-29
> > > > **The basic representation of entity in [1] and [2] has been misunderstood**
> > > >
> > > > It is not correct that „in [1, 2], each entity is represented with a high dimensional ball with parametrized centers and radii learned from data, as mentioned in the review.“
> > > >
> > > > In [2] page 3, it is clearly stated that „Let the decision region for each tag be modeled with a Euclidean ball centered at the embedding point of the tag, so that any image is tagged as, say, “cat” if and only if it is embedded inside the Euclidean ball corresponding to the “cat” tag.“ In [2] page 63, the author continues that „The idea is to ensure that the decision regions for the labels are constrained to match a conceptual Venn diagram that expresses the desired logical constraints between the labels, in terms of their inclusion and exclusion relationships.“
> > > >
> > > > It is clear that in [1] and [2] entities are represented by points, instead of high dimensional balls.
> > > >
> > > > Our work does not stem from [1] and [2], and has a number of fundamental differences.
> > > >
> > > > 1. Different representation framework. (a) Our work is region-based embedding (n-ball). So 'animal' and 'flower' are represented as high-dimensional regions. 'dog is an animal' is represented as the region of dog is inside of the region of animal; [1] and [2] are vector-based embedding, in there work, 'animal' and 'flower' are represented as points. 'dog is an animal' is interpreted as the distance from dog to animal is less than a threshold. (b) our work impose tree-structures ONTO pre-trained vector embeddings; while [1] and [2] introduces decision regions WITHIN the vector space.
> > > >
> > > > 2. Different methods. Our work introduces novel geometric construction into learning representation; [1] and [2] follow the standard method – minimizing some objective functions.
> > > >
> > > > 3. Different quality of imposition. With geometric construction, we achieve precise imposition (zero energy cost). The methods of [1] and [2] cannot.

---

> > > > > ### Comment · AnonReviewer1 · 2018-12-31
> > > > > **Not really...**
> > > > >
> > > > > The critical point of argument is that in [1, 2] corresponding to each entity, a high dimensional ball in the embedding space is learned from data  with a learned center position for each ball as well as a learned radius.  And that the hierarchy relations are incorporated by relations between those balls.
> > > > >
> > > > > In terms of English wording, yes, they may have referred to the whole learned ball as the entity's learned "decision region" and referred to the "center" of that ball as the "embedding location", while you refer to the whole ball as the "embedding". Naming difference..
> > > > > Does not change the close conceptual similarity of the methods and the need citation!
> > > > >
> > > > > In any event, the list of differences and similarities between the super-related methods that you are establishing above is a beneficial thing for readers, missing of which was a little surprising at the beginning.

---

> > > > > > ### Author Response · Authors · 2019-01-01
> > > > > > **Re: Not really...**
> > > > > >
> > > > > >
> > > > > > in [2] page 65, s_k(x) is defined as s(x, y_k = 1) - s(x, y_k = 0)
> > > > > > in page 66, y_1 ==> y_2 is explained as whenever the label y_1 is set to 1 (true) then the label y_2 must also be set to 1 (true).
> > > > > > however, the formalism of y_1 ==> y_2 is defined as  s_1(x) >=  -\delta ==>  s_2(x) >=  \delta
> > > > > >
> > > > > > please explain why  s_1(x) only needs to be greater than  minus \delta (-\delta).

---

> > > > > > > ### Comment · AnonReviewer1 · 2019-01-06
> > > > > > > **Re: Not really...**
> > > > > > >
> > > > > > > Theorem 1 in Section 5.2.2 in page 69 of [2] proves it. The proof of the theorem appears in Appendix C.3 page 98 of [2] and the geometric intuition behind it appears in the second paragraph of Section 5.2.2  and Figure 5.2 in page 70 of [2].
> > > > > > > It is also discussed in Theorem 3 of reference [1].

---

> > > > > > > > ### Author Response · Authors · 2019-01-06
> > > > > > > > **unacceptable comment**
> > > > > > > >
> > > > > > > > please use your own words to do the explanation (why  s_1(x) only needs to be greater than  minus \delta (-\delta))
> > > > > > > >
> > > > > > > > thanks.
> > > > > > > >
> > > > > > > > PS: the theorems you pointed in [1] and [2] do not directly go to the formalism, something is still missing.

---

> > > > > > > > > ### Comment · AnonReviewer1 · 2019-01-06
> > > > > > > > > **Not really...**
> > > > > > > > >
> > > > > > > > > It is not clear which part is exactly under question, one variable implying the other one,  or insertion of deltas, etc?
> > > > > > > > >
> > > > > > > > > The bottomline is that once their constraints are held (i.e. once the ball for y_1 is inside the ball for y_2),  then nonnegative s_1(x)  has to imply nonnegative s_2(x). And inserting margin (\delta) in the proper way in the constraints does not change that.
> > > > > > > > >
> > > > > > > > > When they insert margins around balls, for a y_1=> y_2 implication rule , it is enforced that whenever a point is not only inside the inner ball but even margin away from the boundary of it (but outside),  it has to be not only inside the boundary of the outer ball, but also margin away from the boundary (but inside).  Enforcing such constraint guarantees y_1 implies y_2.  This is to provide separation between the balls and facilitate margin based models.
> > > > > > > > >
> > > > > > > > > This lengthy discussion seems more like a distraction from the main point and out of place here.

---

> > > > > > > > > > ### Author Response · Authors · 2019-01-07
> > > > > > > > > > **exact the point, not a distraction**
> > > > > > > > > >
> > > > > > > > > > if we understand your explanation correctly, the implication rule y_1 ==> y_2 will not be logically expressed in terms of s_1(x) and s_2(x), in both [1] and [2]. The correct way shall be one of the two equivalent assertions:  (1) s_1(x) >= \delta ==> s_2(x) >= \delta, (2)  s_1(x) < -\delta \or s_2(x) >= \delta.

---

> > > > > > > > > > > ### Comment · AnonReviewer1 · 2019-01-07
> > > > > > > > > > > **Re: exact the point, not a distraction**
> > > > > > > > > > >
> > > > > > > > > > > No. The stronger margin-based constraint they enforce is actually:   s_1(x) >= -\delta => s_2(x) >= \delta.
> > > > > > > > > > >
> > > > > > > > > > > Note that delta itself is non-negative i.e. \delta>=0 and remember s_k(x) >=0 means label k is active for input x. Now, to check what happens to x with  s_1(x) >= 0, see below.
> > > > > > > > > > >
> > > > > > > > > > > Suppose s_1(x) >= 0. Since 0 >= -delta, it implies s_1(x) >= -delta, so matches with left hand side (LHS ) of the above rule and it triggers it, and as a result RHS of the above rule has to hold. So it is implied that s_2(x) >= delta >=0.
> > > > > > > > > > >
> > > > > > > > > > > In summary,  s_1(x) >= 0 => s_2(x) >= 0, for all x.  In other words, whenever label 1 is active for x, label 2 would be active for it too.
> > > > > > > > > > >
> > > > > > > > > > > [It is easy to check why the assertions you are suggesting in last comment does not work.  Because it does not give any guarantee for x when 0<= s_1(x) < delta.  The label 1 for this specific x is active, but label 2 would not necessarily be since your suggested rule above is not get triggered.]